# Importance of Optical Coherence Tomography and Optical Coherence Tomography Angiography in the Imaging and Differentiation of Choroidal Melanoma: A Review

**DOI:** 10.3390/cancers14143354

**Published:** 2022-07-10

**Authors:** Iwona Obuchowska, Joanna Konopińska

**Affiliations:** Department of Ophthalmology, Medical University in Bialystok, 15-089 Bialystok, Poland; joannakonopinska@o2.pl

**Keywords:** choroidal melanoma, choroidal nevi, optical coherence tomography, optical coherence tomography angiography, OCT, OCTA

## Abstract

**Simple Summary:**

Choroidal melanoma is a highly malignant intraocular neoplasm. It is the most prevalent intraocular tumor in adults and it derives from melanocytes; the liver is the most common site of its metastases. An early and noninvasive diagnosis is essential to enhance patients’ chances for early treatment. The OCT and OCTA-A are a noninvasive and noncontact methods used in the diagnostic process and support therapeutic decisions during control visits. These devices allow for detection and the real-time imaging of choroidal melanoma and its differentiation from metastasis and choroidal nevi. In addition to analyzing the currently utilized OCT and OCT-A methods, this review describes also the anatomy and imaging of specific vascular layers of the eye.

**Abstract:**

Choroidal melanoma requires reliable and precise clinical examination and diagnosis to differentiate it from benign choroidal nevi. To achieve accurate diagnosis, as well as monitoring the progression of disease, various imaging modalities are used, including non-invasive optical coherence tomography (OCT) and optical coherence tomography angiography (OCTA). This review begins with a historical account of the development of OCT and OCTA and the methods of generation of images. This outlines the understanding of what OCT/OCTA images show, as well as how image artifacts arise. The anatomy and imaging of specific vascular layers of the eye are introduced. Then, anatomical aspects of choroidal melanoma, its diagnosis and differentiation from metastasis, and choroidal nevi are presented. The purpose of this review is to critically evaluate application of OCT and OCTA in the diagnosis of choroidal melanoma.

## 1. Introduction

Choroidal melanoma is a highly malignant tumor and the most common primary intraocular malignancy found in adults. Its early detection and implementation of appropriate treatment are critical to prevent distant spread and prolong patient survival. Fine-needle biopsy (FNAB) is used routinely in clinical practice to perform genetic testing for diagnosis/prognosis purposes and is suitable for establishing the ultimate diagnosis, although it raises the potential risk of cancer-cell spread and retinal damage [1]. This puts emphasis on other diagnostic modalities. Some methods that have been used for years, such as fluorescein angiography or indocyanine green angiography, have multiple drawbacks. In this context, implemented 30 years ago, non-invasive optical coherence tomography (OCT), appears is important and effective diagnostic modality for evaluating choroidal melanoma [1]. OCT seems particularly useful in distinguishing between choroidal melanoma and other retinal or choroidal masses, metastatic tumors, and, most of all, choroidal nevi. Differentiating between a small choroidal melanoma and choroidal nevus still constitutes a challenge. The fact that choroidal melanoma has a dense intrinsic vasculature points to a novel diagnostic technique, optical coherence tomography angiography (OCTA), as a particularly useful tool for evaluating the tumor [2,3]. OCTA is suitable for non-invasive visualization of choroidal melanoma microvasculature and imaging vascular lesions in adjacent tissue. Both OCT and OCTA also seem to be highly effective techniques for monitoring treatment outcomes in choroidal melanoma.

## 2. Optical Coherence Tomography and Optical Coherence Tomography Angiography

OCT and OCTA are optical imaging techniques to visualize eye tissue in real time without exposure to harmful ionizing radiation. OCT was first presented as a novel technique for eye-tissue imaging in 1991 by professor James Fujimoto and colleagues from the Massachusetts Institute of Technology. In the OCT technique, the light from a low-coherence source is used to obtain two- and three-dimensional tissue images with a very high spatial resolution. Since its first presentation, OCT has undergone multiple modifications, becoming an essential diagnostic technique enabling ophthalmologists to examine the eye from a completely different perspective [1].

The working principles of OCT are very similar to medical ultrasound, except that OCT uses light instead of sound. Light travels much faster than ultrasound (in a vacuum at 300,000,000 m/s, while sound travels at approximately 300 m/s). OCT uses the rule of indirect low-coherence interferometry, in which a beam of light is concentrated onto the retina. Part of the light is directed to the sample and another portion is sent to a reference arm with a well-known length. As a result, backscattered light travels to a detector, which is compared to a reference beam of a known length to calculate the echo time delay of light. When the distance between the light source and retinal tissue equals the distance between the light source and reference mirror, the reflected light and the reference mirror interact to produce an interference pattern. The interference measured by the photodetector is then converted to an A-scan signal. Axial imaging depth delimits the axial range, which is covered in a B-scan. It is defined by the maximum fringe frequency that can be identified, because maximum frequency of the interference spectrum decodes the maximum depth [4].

Unlike conventional fundoscopy, OCT provides information about the depth at which a pathological lesion is located within the eye. An OCT image resembles a histological cross-section of the retina, with all its layers visible clearly. In this method, light-refractive indices are analyzed across various retinal layers to obtain high-resolution scans of the retina. In the initial technique, the source of light in OCT was a superluminescent diode with a wavelength of 820–840 nm; nowadays, we can find OCT systems based on tunable laser sources as well. Individual axial OCT scans, known as A-scans, document the reflectivity of the tissue along the light beam scanning the retina at various depths. Multiple A-scans recorded during a single examination are combined into a tomogram, referred to as a B-scan. In this context, OCT imaging shares some analogies with ultrasonography, except that it utilizes light waves instead of high-frequency sound waves, and the optical technique provides higher resolution and does not require direct contact with examined tissue. OCT images have a high axial resolution, 3–5 µm in tissue depending on the light source, and high transverse resolution. The depth of the scan is 2 mm, whereas its width varies from 4 mm to 10 mm. Unfortunately, the image’s resolution and signal intensity decrease with the depth of the examined tissue [2].

Individual retinal layers on OCT scans are identified as alternating high-reflective bands (HRB) and low-reflective bands (LRB). The order of the bands and the reflectivity thereof, starting at the level of the corpus vitreum, is as follows: nerve fiber layer (HRB), ganglion cell layer (LRB), inner plexiform layer (HRB), inner nuclear layer (LRB), outer plexiform layer (HRB), outer nuclear layer (LRB), external limiting membrane (HRB), myoid zone (LRB), ellipsoid zone (EZ) (HRB), outer segment of photoreceptors (LRB), interdigitation zone (IDZ), and retinal pigment epithelium–Bruch’s membrane complex (HRB). EZ represents a junction between the inner and outer segments of photoreceptors, or, as postulated by some authors, corresponds to the anatomical location of the ellipsoid portion of the inner segments of photoreceptors. The myoid zone is not a proper retinal layer, but contains ribosomes, endoplasmic reticulum, Golgi bodies, and mitochondria. IDZ corresponds to the tips of outer photoreceptor segments [4,5,6]. More deeply located are the choroid and the sclera, both poorly visible on conventional OCT scans.

There are two main technologies of OCT: (1) time-domain OCT (TD-OCT), an older technology in which the information is collected as a function of time, and (2) Fourier domain or frequency domain OCT (FD-OCT), a more modern technology based on measurement in the frequency domain using Fourier transform. The foundation of TD-OCT is the time-of-flight measurement of a light wave reflected from examined tissue. The light emitted by a semiconductor diode is directed onto the optic bundle separator, which splits it into two portions. One portion is directed onto the reference mirror and the other on the examined tissue, whereby it is reflected by the tissue structures. The light beams reflected by the examined tissue and the reference mirror are superimposed. Whenever the optical routes of the beams are equal, light-wave interference occurs. The reference mirror in TD-OCT is mobile, and as a result, neither the image quality nor the scanning speed is satisfactory. Those drawbacks were eliminated in the FD-OCT technology by applying a static mirror and using a partially coherent light source with detection in the optical frequency domain, rather than in the time domain. As a result, scanning speed in FD-OCT can be up to 100-fold higher than in TD-OCT [3,7].

FD-OCT technology is used in two types of devices: (1) spectral-domain OCT (SD-OCT) scanners using a spectrometer for the detection of an interferometric signal, and (2) optical tomography scanners using a tunable laser source and a photodetector instead of the spectrometer—the so-called swept-source OCT (SS-OCT) devices. Conventional SD-OCT is the most frequently used OCT technique nowadays, suitable for obtaining cross-sectional retinal images with an axial resolution of up to 5 µm [8]. The advantages of this method include a very high scanning speed suitable for obtaining up to 40,000 A-scans per second, a high number of images per unit area, good visualization of individual retinal layers, and a possibility to analyze retinal morphology quantitatively and perform its three-dimensional reconstruction, vital for vitreoretinal surgery planning [9]. An even more advanced technique is SS-OCT, which utilizes longer waves from the infrared spectrum (>1000 nm). Such waves penetrate deeper into the tissue and eliminate shadowing artifacts, e.g., from blood vessels. Additionally, longer waves penetrate better across lens opacifications and hemorrhages. Due to the use of longer waves and replacement of the sensitivity-limiting spectrometer for a photodetector, SS-OCT scanners have many advantages, such as higher scanning speed, reduced sensitivity loss, better penetration of light and capturing of signals from deeper layers, and denser scan patterns in a larger scanning area [10]. An algorithmic modification of SS-OCT contributed to reducing the granular noise signal and enabled obtaining the so-called “speckle noise-free” images [11].

Despite many advantages, conventional SD-OCT also has some limitations, especially in evaluating deeper structures, such as the choroid [9]. The limitations stem from the fact that the light from the close infrared spectrum is dispersed by the retinal pigment epithelium, and as a result, the returning reflection from the choroid is very weak. Although devices using light with longer wavelengths can visualize deeper structures, the resolution of the images is lower, which limits their application in clinical practice.

New diagnostic perspectives appeared in 2008, when due to the innovation of SD-OCT technology, a new imaging method, enhanced depth imaging OCT (EDI-OCT), was obtained. The new technology uses light with 840 nm wavelength. In this method, the imaging focal point is moved more posteriorly, so the largest portion of the light beam is focused at the choroid and internal sclera level, enabling more accurate visualization of those structures. In EDI-OCT, the examined object is closer to the device, resulting in lower delay and higher resolution of the scans. The averaged image is synchronized with a real-time eye-tracking system that markedly reduces motion and noise artifacts [12,13]. A side effect of moving the imaging focal point deeper inside the eye to better visualize the choroid is less accurate imaging of the corpus vitreum. Therefore, in SD-OCT devices, the operator can choose between two imaging modes: a conventional scan visualizing the corpus vitreum and retina and an EDI mode to visualize the retina and choroid. In most cases, it is not possible to accurately visualize such two distant structures as the corpus vitreum and choroid simultaneously.

The most advanced OCT technique available nowadays is ultrahigh-resolution OCT (UHR-OCT), with a resolution of 1 µm, unheard of in the case of other ophthalmological imaging methods. This UHR-OCT technology is currently used solely for research purposes, but hopefully will soon be available commercially [14]. To put the advantages of UHR-OCT in perspective, average resolutions of ultrasonography (US), magnetic resonance imaging (MRI) and computed tomography are 150–450 µm, 650–800 µm, and 500–625 µm, respectively. Compared with those figures, even conventional OCT provides images with an excellent resolution.

The introduction of OCT angiography in 2014 constituted another milestone in ophthalmological diagnosis. OCTA is a novel technique of eye imaging, enabling in vivo structural analysis combined with analysis of blood flow through retinal vessels and choroid capillaries without using a contrast agent. OCTA combines the advantages of tissue imaging (OCT) and microcirculation assessment (angiography). The foundation of OCTA is SSADA (split-spectrum amplitude-decorrelation angiography) technology, in which blood vessels are identified based on the detection and measurement of blood flow using the reflection of light from the surface of circulating erythrocytes. This enables the assessment of the following anatomical and vascular structures, starting at the level of the corpus vitreum: (1) superficial capillary plexus (SCP; vessels in the nerve fiber layer of the retina), (2) deep capillary plexus (DCP, vessels at the level of the inner nuclear layer and outer plexiform layer of the retina), (3) outer retina, and (4) choriocapillaris. OCTA can detect vascular pathologies in all those layers, examined simultaneously or analyzed separately [15,16].

The first OCTA devices were suitable for examining 2 × 2 mm or 3 × 3 mm areas, which limited their application solely to the evaluation of the macula. Thanks to the development of novel algorithms, OCTA can now be used for evaluating larger areas: 12 × 12 mm or 15 × 9 mm [17]. Recent evidence suggests that the field of view of OCTA can be extended even further, due to the implementation of additional convex lenses. In future, extended-field images of OCTA (EFI-OCTA) might be used to evaluate the peripheral retina and choroid. To date, the largest field of view can be obtained with 102^0^ fluorescein/indocyanine green angiography (FA/ICGA); however, extended-field image swept-source OCTA (EFI-SS-OCTA) ranks second among currently available optical techniques for ophthalmological examination, being superior to conventional 55^0^ FA/ICGA and OCTA without EFI [18]. The features of various types of OCT systems have been shown in the Table 1.

## 3. Choroidal Melanoma

Choroidal melanoma is a highly malignant intraocular neoplasm occurring in adults. It is derived from melanocytes of the choroid, the mid-layer of the eye’s wall, responsible primarily for supplying blood to the retina located above. Approximately 98% of choroidal melanomas occur in Caucasians, with the global prevalence of this malignancy in people of this heritage estimated at 5.1 per million [19]. The prevalence of choroidal melanoma in Caucasians from northern Europe is higher than in the global population, approximately 8 per million, compared with 5.1 to 6 per million in the United States, 3.3 per million in Italy, and 1.9 per million in Spain [18,19,20]. The prevalence of this malignancy in other populations, including non-Caucasians, i.e., Native Americans, Africans, Asians, Middle Easterners, and Asian Indians, is lower than 1% [21]. The mean age at diagnosis of choroidal melanoma is 58–60 years, with the malignancy slightly more often found in men than in women (51%) [22,23]. When stratified according to sex, the prevalence of choroidal melanoma is estimated at 4.9 per million in men and 3.7 per million in women [24]. Only 1% of choroidal melanomas are diagnosed in patients younger than 20 years, and the remaining 99% are found in persons ≥21 years of age. Choroidal melanoma is virtually never detected in children.

Manifestations of choroidal melanoma depend on the location of the tumor. More peripheral tumors located outside the visual axis can be asymptomatic, as are the small tumors. Lesions located closer to the posterior pole of the globe, larger tumors, or masses with concomitant exudative retinal detachment produce multiple manifestations, including flashes, floaters, visual acuity loss, or visual field defects. The type of complaints reported by melanoma patients also depends on the tumor configuration. Choroidal melanoma may have the form of a dome (75%), fungus (20%), or present as a diffuse lesion (5%) [19]. Approximately 55% of choroidal melanomas are pigmented; non-pigmented and mixed tumors represent 15% and 30% of the lesions, respectively [21]. The mean diameter of the tumor base at the time of diagnosis is 11.3 mm, with a mean thickness of 4–5 mm [19,25]. Depending on their thickness, choroidal melanomas are classified as small (0–3.0 mm), medium-sized (3.1–8.0 mm), and large tumors (≥8.1 mm). A widely accepted classification of choroidal melanoma proposed by the American Joint Committee on Cancer (AJCC) includes four stages, from the least advanced T1 to T2, T3, and T4, with a large number of subcategories depending on the involvement of the ciliary body and extraocular spread of the tumor [26]. According to the literature, choroidal melanomas are most often diagnosed as stage T1 tumors, and T4 lesions are rarely found [27]. Clinical characteristics of choroidal melanomas of various stages are summarized in Table 2.

### T-Tumor

The less advanced the disease, the better the prognosis. Choroidal melanoma may form distant metastases, most often to the liver (91%), lungs (28%), bones (18%), and skin (12%) [28,29]. The evidence from clinical studies suggests that the risk of extraocular spread is the lowest in the case of small melanomas with a thickness no greater than 1 mm [30]. This fact puts particular emphasis on appropriate diagnosis and early detection of the tumor as prerequisites for its effective treatment.

Choroidal melanoma is typically detected during routine slit-lamp biomicroscopy and/or indirect fundus ophthalmoscopy under the dilated pupil. If an intraocular tumor is detected, further management is slightly different from that used in the case of neoplasms in other locations. Even if a mass is considered malignant, which is undoubtedly the case for choroidal melanoma, no biopsy specimens for histopathological evaluation are taken so as to avoid iatrogenic tumor spread and/or retinal laceration. However, this does not mean that biopsies of choroidal melanomas are not performed at all. According to the new diagnostic algorithm, a biopsy can be performed at the time of initial diagnosis. The reason for the biopsy, however, is not to confirm the melanoma diagnosis, as this goal can be achieved with specialized ophthalmological examinations, discussed in detail below. Instead, the biopsy is carried out to obtain material for genetic testing, based on which a patient-specific risk of metastasis to other organs can be estimated and an optimal therapeutic strategy can be selected [31]. There are some published reports on real-time (4D-MIOCT) 27-gauge transvitreal biopsy of choroidal melanomas conducted intraoperatively under the guidance of so-called microscope-integrated OCT (MIOCT) [32]. An advantage of this method is the possibility of obtaining a sample from the appropriate depth of the tumor, with reduced risk of injury to the neurosensory retina or excessively deep biopsy with resultant intra-choroidal hemorrhage. However, despite the attempts to incorporate biopsy into the evaluation of choroidal melanoma, this procedure is not performed routinely, given its potential adverse effects and flaws, which are not counterweighed by clinical benefits.

Thus, purely ophthalmological evaluation becomes vital for confirming that a given intraocular mass is indeed malignant. The armamentarium of the ophthalmological methods includes non-invasive techniques, such as ultrasonographic examination (B-scan US or color Doppler) and OCT, and invasive tests involving the use of contrast agents, namely, FA and ICGA. However, the latter two have been recently replaced by non-invasive OCTA. CT and MRI play less important roles in the evaluation of intraocular masses. Those imaging methods primarily find application in the evaluation of the extraocular expansion of the tumor, as well as in differentiating between contrast-enhanced choroidal melanoma, non-contrast-enhanced choroidal detachment, and highly calcified choroidal osteoma [20]. CT and MRI can also be helpful in evaluating masses located in opaque optical centers (scleral leukoma, cataract, intravitreal hemorrhage), in the case of which no ophthalmological methods can be applied other than ultrasonography.

In patients in whom choroidal melanoma is suspected, the diagnostic method of choice is still the 10 MHz B-scan US. This diagnostic modality can detect tumors with thicknesses of >1.0 mm, even in eyes with opaque optical centers [33]. On B-scans, choroidal tumors appear as hyperechoic masses of various shapes, with a choroidal excavation and sometimes with concomitant secondary retinal detachment. US is also suitable for the measurement of the tumor. FA plays a minor role in the case of small tumors. In patients with larger masses, this method shows a characteristically patchy fluorescent pattern associated with the dye leaking from the RPE. Intrinsic vasculature of the tumor, described as a “double-circulation pattern,” less visible on FA, can be observed better on ICGA. On angiographic images, pigmented tumors present as moderate-degree hypoautofluorescence, whereas amelanotic tumors appear as hyperautofluorescence of a moderate degree [34]. The diagnosis can be confirmed on color Doppler, showing typical pulsatile blood flow at the base of the melanoma.

Does the availability of the diagnostic modalities described above justify research on alternative methods? Could OCT and OCTA add anything new to the evaluation of choroidal melanomas? Answers to those questions can be found in further sections of this paper.

## 4. Choroidal Melanoma in Optical Coherence Tomography

OCT is not a method of first choice in patients with suspected choroidal melanoma. Older OCT techniques, including SD-OCT, have limited application in evaluating the choroid, as they poorly visualize this structure; however, they are helpful in detecting concomitant lesions in the retina. SD-OCT can visualize subretinal fluid; as such, it can be helpful in the detection of exudative retinal detachment, a common comorbidity in patients with choroidal melanoma [35,36,37]. However, a more modern technique, EDI-OCT, due to deeper penetration of the light beam, can visualize melanoma as choroidal shadowing, the degree of which depends on the pigmentation of the malignancy [23]. EDI-CT also enables the measurement of the lesion more accurately than ultrasonography. Shields et al. demonstrated statistically significant differences in the sizes of the same tumors measured on US and EDI-OCT [38]. The authors analyzed small choroidal melanoma, and found that tumors with a mean thickness of 2300 µm (2.3 mm, range 1700–3400 µm) on US had a mean thickness of 1025 µm (range 639–1410 µm) when measured on OCT. The measurement error was 55%, and turned out to be statistically significant. Those findings imply that ultrasonographic measurements are inaccurate and overestimate the size of choroidal melanomas. The differences might result from the inadvertent inclusion of flat overlying retina and sclera in the tumor mass. Accurate identification of tumor margins on US and differentiating between the choroid and retina in the anterior segment of the tumor or sclera in its posterior segment constitutes a challenge. Given its resolution, US has substantial limitations in measuring ocular microstructures; such limitations do not exist in the case of OCT. The measurement error for EDI-OCT is only 5–10 µm compared with up to 200–400 µm for ultrasonography [36]. Another study comparing the size of choroidal nevi measured with US and OCT showed that the thickness of the nevi measured on EDI-OCT was 54% lower than on ultrasonography [39]. According to Torres et al. [40], EDI-OCT measurement is easier in the case of smaller tumors, with diameters at the base of <9 mm and thickness of <1 mm, and even the melanomas being too small to be visualized on US can be easily observed and measured on OCT. Ultrasonography is more useful in the case of large and thick choroidal melanomas, as the margins of such tumors extend beyond the area visible on the EDI-OCT scan.

On EDI-OCT images, choroidal melanomas have homogeneous optical reflectivity along the anterior surface, with shadowing throughout deeper layers of the tumors [38]. The pigment epithelium covering the tumor is atrophic or absent. Other characteristic features of melanoma include the loss of photoreceptors and compression and thinning of the choriocapillaris. The latter feature is found in 100% of tumors. In some cases, a rupture of Bruch’s membrane can be seen, particularly in large melanomas. Also, deposits of lipofuscin can be found in the retina. Lipofuscin is a product of cellular metabolism and a natural marker of intracellular processes occurring in photoreceptors and RPE [41]. The analysis of lipofuscin deposits provides important information about the status of the retina, including its most critical layer, photoreceptors. Lipofuscin is a fluorophore and has fluorescence potential, i.e., absorbs light at a specific wavelength and emits light at another wavelength. This feature of lipofuscin has been utilized in ophthalmological evaluation in the technique known as fundus autofluorescence (FAF). Autofluorescence, i.e., the emission of light by lipofuscin, facilitates its detection. At the fundus, lipofuscin is visible as orange pigment on indirect ophthalmoscopy. Lipofuscin deposits are often found concomitantly to choroidal melanoma. While the presence of the orange pigment is not considered a pathognomic sign of choroidal melanoma, lipofuscin is only occasionally found in the RPE and retina adjacent to other choroidal pathologies, such as choroidal nevi, choroidal hemangioma, or choroidal metastasis [42]. FAF allows for the visualization of even subclinical lipofuscin deposits, which highlights the role of this test in the early detection of choroidal melanomas, especially with small diameters, and differentiating between the melanomas and choroidal nevi. It must be stressed that the presence of the orange pigment belongs to the most important prognostic factors associated with the malignant transformation of melanocytic lesions of the choroid into melanoma [43]. However, an increase in the intensity of the FAF signal on the tumor surface does not necessarily correspond to the presence of the orange pigment, as the latter is not the only source of FAF. In some cases, autofluorescence may be associated with pigmentation, sub-RPE drusen, or fibrous metaplasia of RPE [44]. Thus, the correct interpretation of this phenomenon is of utmost importance.

Careful evaluation of the retina on EDI-OCT can show a plethora of changes in its layers, characteristic already for small melanomas and absent in pigment nevi. The presence of such changes in the retina should raise suspicion of a choroidal malignancy. The changes found on EDI-OCT include abnormal or shaggy photoreceptors, loss of external limiting membrane, loss of the inner segment–outer segment junction (loss of EZ), abnormality of inner plexiform layer, and irregularity of ganglion cell layer [38]. The term “shaggy photoreceptors” corresponds to their irregularity and presumably elongation of the rods and cones, associated with the presence of freshly accumulated subretinal fluid. According to various estimates, subretinal fluid accumulation occurs in 76–91% of choroidal melanomas [44,45]. From a morphological perspective, shaggy photoreceptors correspond to subretinal macrophages accumulated on the posterior surface of the retina. The presence of subretinal macrophages is described in 49% of small choroidal melanomas [38]. Intraretinal edema of the overlying retina occurs in 48.3% of cases [44]. None of the diagnostic modalities described above can visualize abnormalities in the specific layers of the retina associated with the presence of choroidal melanoma as accurately as OCT.

## 5. Choroidal Melanoma or Nevus: Differential Diagnosis

A choroidal nevus is the most common benign intraocular mass, found primarily in Caucasians. Although the nevus is benign and, in most cases, a non-harmful lesion, it may undergo transformation into melanoma. In the Blue Mountains Eye Study [46], the prevalence of choroidal nevi in the white population was estimated at approximately 7%. The nevi typically present as well-demarcated, flat, round, or longitudinal dark areas of enhanced choroidal pigmentation. However, amelanotic nevi can also occur. Up to 91% of choroidal nevi are localized posteriorly from the globe’s equator with the same frequency, regardless of the quadrant [47]. The mean diameter of the lesion is 5 mm, with a mean thickness of 1–1.5 mm. The nevi are frequently accompanied by drusen and altered RPE [48].

Each choroidal nevus should be routinely monitored for its potential malignant transformation. Early detection of the transformation is critical for the prompt implementation of treatment and improvement of the prognosis. Hence, the identification of risk factors for the transformation of the nevus into choroidal melanoma becomes of vital importance. The list of features that may be associated with increased risk of the malignant transformation includes thickness of the nevus >2 mm, presence of subretinal fluid (cap over nevus or fluid ≤3 mm from nevus margin), accumulation of the orange pigment, close proximity of the fovea or optic disk (<3 mm), ultrasonographic acoustic hollowness, tumor diameter of 5 mm on photography, floaters, flashes, and blurred vision [43,48,49]. In the case of nevi presenting with at least three of the features mentioned above, the risk of malignant transformation during the next 5 years is estimated at 50% [50]. Annual rate of malignant transformation of choroidal nevi in the white population accounts for 1/8.845 [51].

Therefore, differentiating between the nevus and an atypical malignant lesion, differential diagnosis of nevi and very small choroidal melanomas and monitoring of benign masses for their potential malignant transformation constitute a serious challenge. An optimal instrument for those purposes is EDI-OCT, which can detect small lesions not visible on US. Characteristic features of nevi on EDI-OCT include thinning of the choriocapillaris over the nevus (found on 94% of the scans), loss of RPE (43%), loss of photoreceptors (43%), choroidal shadowing deep to the nevus (partial in 59% and complete in 35%), and irregularity of the EZ (37%) [39]. The loss of photoreceptors manifests as their “cleft” or “stalactite” appearance resulting from the prolonged accumulation of subretinal fluid [48]. Such alterations of photoreceptors are observed more frequently in the nevi than in choroidal melanomas [38]. In turn, choroidal nevi do not contain shaggy photoreceptors, a characteristic feature of small choroidal melanomas.

Bruch’s membrane remains intact in 100% of choroidal nevi. In 18% of the nevi, EDI-CT may show an irregularity of the external limiting membrane, and in 6–8%, irregularities in other retinal layers: outer nuclear layer, outer plexiform layer, and inner nuclear layer. However, no abnormalities are seen in the inner plexiform layer, ganglion cell layer, or nerve fiber layer. This implies that the more distant the retinal layer from the choroid, the lesser the risk of its damage. As mentioned above, the three most inner layers of the retina show no abnormalities, whereas the prevalence of alterations within the RPE and photoreceptor layer, located in the closest proximity of the choroid and hence closest to the lesion, is estimated at 43%. Other pathologies associated with choroidal nevi that can be found on EDI-OCT scans include drusen (41%), accumulation of subretinal fluid (16%), presence of the orange pigment (6%), non-pigmented hallo (14%) and pigmented hallo (8%) [39]. Detection of such pathologies is crucial in the context of malignant transformation of the nevus. Up to 84.5% of choroidal nevi are pigmented lesions, with non-pigmented and mixed nevi representing 8% each. The only significant difference between pigmented and non-pigmented nevi on EDI-OCT is the more complete choroidal shadowing of the former. Abnormalities of the RPE lining choroidal nevi include atrophy (43%), RPE loss (14%), RPE nodularity (8%), fibrous metaplasia (8%), hyperplasia (4%), and—found markedly less often—RPE detachment and RPE trough.

EDI-OCT can also detect an interesting phenomenon characteristic for a specific variant of choroidal nevus, the so-called halo nevus. This phenomenon, described as “posterior scleral bowing,” is a posterior deflection of the sclera around the base of the nevus [52]. Posterior scleral bowing is associated with the fact that the nevus grows posteriorly, “pushing” the sclera backwards.

The evidence from older studies shows that less technically advanced TD-OCT scanners were not suitable for the accurate assessment of the characteristic features of the nevus, as they visualized only a superficial part of the choroid. Nevertheless, even the older-generation scanners are useful in visualizing RPE and retinal abnormalities coexisting with choroidal nevi. Shields et al. [53] used TD-OCT to demonstrate the following associated pathologies occurring concomitantly with 120 choroidal nevi: decrease in photoreceptor density or photoreceptor loss (51%), drusen (41%), subretinal fluid (26%), retinal thinning (22%), retinal edema (15%), and RPE detachment (12%). In turn, Muscat et al. [36], using TD-OCT to evaluate retinal abnormalities accompanying choroidal nevi, demonstrated the presence of subretinal fluid in 30% of patients and changes in the structure of the retina lining the nevus in 8%.

Aside from visualizing choroidal nevi and associated retinal and choroidal pathologies, EDI-OCT plays an essential role in the accurate measurement of the masses and determining their topography in relation to the macula and optic disk. Those parameters are important for assessing the risk of malignant transformation. The mean maximum diameter of the nevus base is 4.2 mm (median 4.0 mm, range 0.5–11.0 mm), and the thickness of most nevi does not exceed 1.5 mm. An increase in nevus thickness is usually a marker of its malignant transformation. The mean distance between the nevus and the fovea is 2.7 mm (median 2.0 mm, range 0.0–10.0 mm), whereas the mean distance to the optic disk is estimated at 3.2 mm (median 3.0 mm, range 0.0–13.0 mm) [39]. The accuracy of the EDI-OCT measurements of the nevi is much higher than in the case of US, until recently the only available method to measure the intraocular masses. US-based measurements of choroidal nevi were shown to be overestimated by 126% [39], with the measurement error being much higher than in the case of choroidal melanomas. The discrepancies between the EDI-OCT- and US-based measurements of choroidal nevi may be associated with their smallness, especially in terms of thickness. Many choroidal nevi are too flat to be accurately measured on US. Another potential reason behind the inaccuracy of US-based measurement may be the previously mentioned phenomenon of posterior scleral bowing [52].

The usefulness of ophthalmoscopy, US, and EDI-OCT in detecting subretinal fluid accumulation accompanying choroidal nevi differs considerably. Accumulation of subretinal fluid is virtually not visible on US, and the effectiveness of ophthalmoscopy in detecting subretinal fluid deposits is approximately 50% lower than OCT [39,53]. Importantly, the presence of subretinal fluid is an established risk factor for the malignant transformation of choroidal nevi.

## 6. Choroidal Melanoma or Metastasis: Differential Diagnosis

Metastases to the choroid are the second most common intraocular malignancies after melanomas. Breast, lung, and prostate cancers are primary tumors that most often metastasize to the choroid. Metastases to the fundus usually present as yellowish masses located below the retina, often with subretinal fluid on their surface [54,55]. Most metastases are small or medium-sized masses, with a mean thickness of 3.0 mm. They are frequently localized around the equator or in the posterior pole, and due to such location, can be visualized on OCT [56]. EDI-OCT can detect otherwise easy-to-overlook subclinical choroidal metastases, barely visible on ophthalmoscopy [57]. A majority of metastatic tumors found in the choroid are flat (75%), with the remaining 25% being convex. On EDI-OCT scans, metastases present as partial (71%) or complete (21%) choroidal shadowing. The shadowing is not observed in merely 8% of the patients [58]. Up to 71% of the metastases have low optical reflectivity. According to Al-Dahmash et al. [58], the most characteristic OCT features of metastases to the choroid include “lumpy bumpy” anterior surface (64%), compression of the underlying choriocapillaris (93%), and posterior shadowing (86%).

Thinning of the overlying choriocapillaris (100%), shaggy photoreceptors (79%), subretinal fluid (75%), and thinning/atrophy of RPE (53%) belong to abnormalities associated with the presence of metastatic tumors, frequently observed on EDI-OCT. In another study, pathologies of overlying RPE were found in 78% of eyes with metastatic tumors. Other—more or less often observed—abnormalities include the atrophy or thinning of Bruch’s membrane (25%), pathologies in the outer segments of photoreceptors (27%), loss of interdigitation in the outer segments of the cones (64%), and abnormalities of the external limiting membrane (29–50%), outer nuclear layer (27%), outer plexiform layer (23%), and four most superficial retinal layers (inner nuclear layer, inner plexiform layer, ganglion cell layer, and nerve fiber layer, 9% each) [58,59]. In the study conducted by Al-Dahmash et al. [58], no pathologies were found in the four terminal layers of the retina; in 79% of the cases, the presence of metastatic tumors was associated with the accumulation of subretinal fluid, and lipofuscin deposits and intraretinal edema were observed in 7% and 14% of cases, respectively. The mean thickness of metastatic tumors on EDI-OCT is 854 µm (range 287–1500 µm), with ultrasonographic measurements being overestimated by 59% [5].

An important difference in the EDI-OCT presentation of choroidal melanomas and choroidal metastases is that most of the latter have a plateau-shape configuration, while dome-shaped elevation is a predominant configuration of the melanomas [38]. Another distinctive feature is the tumor surface, smooth in the case of choroidal nevi and melanomas and rather irregular in the case of metastatic lesions [59]. Accumulation of subretinal fluid is equally frequent in choroidal metastases (67–100%) and choroidal melanomas (76–100%) [44,55,58,59]. Recent studies have identified the following differences in the EDI-OCT presentation of choroidal melanomas and metastatic tumors: the anterior surface of the choroid is regular and smooth in 89.7% of melanoma eyes and 47.6% of eyes with metastatic lesions, and the anterior contour of the tumor is lobulated in 81% of metastases and only 17.2% of melanomas [44]. Interestingly, although metastases to the choroid may originate from various primaries, their presentation on EDI-OCT scans is similar, regardless of the origin.

OCT is not an appropriate diagnostic instrument in the case of peripheral and very large tumors, as the periphery of the retina and choroid is barely accessible with this imaging modality [59]. However, this problem may be solved soon, as SS-OCT technology with 20- and 40-diopter indirect ophthalmoscopy lenses can provide wide field-of-view images of the peripheral tumors [60]. A wide field-of-view image corresponds to a scanning area equal to 4.6 times the distance between the optic disk and fovea; meanwhile, the mean scanning area accessible with SD-OCT is only 2.1 times the distance between the two structures mentioned above.

## 7. Choroidal Melanoma and Choroidal Nevi on Optical Coherence Tomography Angiography

Since choroidal melanomas are well-vascularized lesions, OCTA has high diagnostic value, especially in differentiating between malignant and benign intraocular masses.

On OCTA images, melanocytic lesions present heterogenic (61.4%) and hyperreflective (81.8%) in the case of choroidal nevi and isoreflective or hyporeflective (62.5%) with a hyperreflective ring in the choriocapillaris layer surrounding the tumor (62.5%) in the case of choroidal melanomas [61,62].

According to Fuste et al. [61], the primary difference between choroidal melanomas and nevi on OCTA images is a well-demarcated border found in 78% of the benign lesions and a blurred border observed in 72% of the malignancies. Other distinctive features include hyperreflective capillary vasculature (85%) and rare avascular areas (17%) for the nevi, and multiple avascular areas (78%), hyporeflective choroidal capillary vasculature (72%), and vascular networks and loops (45%) for the melanomas.

Choroidal nevi have normal, intact RPE–Bruch’s membrane complex and normal outer retinal layers [63]. In the choriocapillaris, the normal homogeneous vascular mosaic is blurred in the area corresponding to the nevus [64]. On OCTA scans, flat nevi present as non-perfused areas surrounded by an intense ring of the microvasculature. The microvascular flow rate per 1 mm^2^ above the lesion at the level of the choriocapillaris is 63.68% (range 60.42–67.62%), being similar to the flow rate in the normal contralateral eye at the same level. In the case of choroidal melanoma, the microvascular flow rate at the level of choriocapillaris overlying the tumor is 55.73% (range 41.93–60.82%), which is a lower value compared with either corresponding location in the contralateral eye or eye with the nevus. Typical abnormalities found on OCTA scans over choroidal melanoma also include deranged and thickened EZ, disrupted and dome-shaped excrescences RPE–Bruch’s membrane complex, and obscured or thickened IDZ [63].

Comparative analysis of some OCTA parameters of the retina in eyes with choroidal melanomas and choroidal nevi allows for identifying features that might be considered distinctive characteristics of benign and malignant lesions in future. Eye with choroidal nevi and normal contralateral eyes have similar central macular thickness (CMT), superficial and deep foveal avascular zone (sFAZ and dFAZ) areas, and superficial and deep capillary vascular density (sCVD and dCVD). Meanwhile, eyes with choroidal melanomas have higher CMT and dFAZ and lower sCVD and dCVD than normal eyes [64]. The decrease in CVD is associated with the presence of subretinal fluid and an increase in tumor thickness. According to Chien et al. [65], dCVD may be the most significant factor differentiating between choroidal melanomas and choroidal nevi. Melanomas cause a decrease in dCVD to induce ischemia, which in turn stimulates the synthesis of vascular endothelial growth factor A. The latter factor promotes vascularization of the tumor, abundant in the case of choroidal melanomas. Some authors have analyzed vascular and perfusion density in the macula in patients with choroidal melanomas. Vascular density (VD) is defined as the total length of red-cell-perfused vasculature per area in a region of measurement, and perfusion density (PD) corresponds to the total area of perfused vasculature per unit area in a region of measurement. In melanoma eyes, both VD and PD in the macula are significantly lower than in normal contralateral eyes [66].

OCTA accurately visualizes the vasculature of choroidal melanomas, both mural and piercing feeding vessels. Choroidal melanomas are characterized by dense and disorganized intrinsic vascularity. Blood vessels, with large and uneven thickness and contorted [48], form rings and small loops. Vessels inside the tumor can be blurred due to the accumulation of pigment, but still visible because of their high density. Naroev et al. [67], who used OCTA to analyze tumor vessels in small choroidal melanomas, found a neovascular component under RPE with a marginal avascular zone corresponding to the tumor slope. The vessels formed loops with multiple turns and anastomoses. Determining tumor vasculature has significant prognostic value, as an increase in vascularity is a marker of malignant transformation. OCTA is superior to FA/ICGA in the visualization of tumor vasculature, as the latter two methods are invasive and pose a risk of complications associated with intravenous administration of a contrast agent. Moreover, some patients are allergic to a dye or have a concomitant disorder of the kidneys, organs through which the dye is eliminated from the body. Finally, intensive pigmentation of melanomas and the presence of small fundal hemorrhages limit the visibility of tumor vasculature on conventional angiography. Meanwhile, OCTA is free from such limitations [68].

Greig et al. [68] used SS-OCTA to analyze the vascular patterns of small choroidal melanomas, elevated choroidal nevi, and flat choroidal nevi. They found choroidal vessels in 100% of melanomas, 77% of elevated nevi, and 17% of flat nevi. All nevi had normal choroidal vasculature with straight and parallel vessels. Central vessels were less visible than the peripheral vessels. The vessels of the nevi had small diameters. Small avascular areas were found in 63% of elevated nevi, but none of the flat nevi. All choroidal melanomas contained avascular areas larger than in the elevated nevi. Vascular patterns varied, with prevalence of multiple vascular loops and crossing vessels. In choroidal melanomas, blood vessels were located deeper than in the nevi. Flow in the choriocapillaris of melanoma eyes was reduced compared with the eyes with flat nevi. However, no differences were found in the choriocapillaris flow within the eyes with elevated nevi and melanoma eyes [69]. The decrease in choriocapillaris flow is associated with compression of this layer and increased lesion depth [62,69].

OCTA has some limitations when it comes to imaging deeper retinal layers, as it only accurately visualizes choriocapillaris closest to the retina. Thus, visualizing deeper medium-sized and large choroidal vessels still constitutes a challenge. Therefore, a standard method to visualize such vessels is still ICGA, which unfortunately is an invasive procedure, requires dilatation of the pupil, and takes much longer than OCTA. A solution to this problem might be the state-of-the-art SS-OCTA, a technology suitable for visualizing deeper vessels of the retina, providing an image quality similar to that of ICGA [70,71].

SS-OCTA has also been used for the differential diagnosis of choroidal melanomas and other masses, such as nevi, circumscribed choroidal hemangioma, optic disk melanocytoma, retinal astrocytic hamartoma, or metastasis [71,72]. Choroidal nevi, retinal hamartoma, and circumscribed choroidal hemangioma are the only lesions with well-demarcated borders. Nevi, hemangioma, astrocytic hamartoma, and melanocytoma have hyperreflective internal structure, not found in melanoma. Unlike nevi and hemangioma, choroidal melanoma can spread onto external retinal layers. Intrinsic tumor vessels are found solely in melanoma and choroidal hemangioma.

Similarly to OCT, OCTA also has some limitations. OCTA is not a preferred method and has a lower diagnostic value in the case of opaque optical centers, large accumulations of subretinal fluid, peripheral tumors or masses with thickness >3.5 mm. Under such circumstances, OCTA scans can be difficult to obtain or have poor quality.

## 8. Monitoring of Treatment Outcomes in Choroidal Melanoma

Treatment of choroidal melanoma depends on its size and location within the eye, concomitant complications, visual acuity in the affected eye, status of the contralateral eye, age, and overall patient condition. Available treatment methods include radiotherapy (brachytherapy, irradiation with charged elementary particles), transpupillary thermotherapy, endoresection, exoresection (transscleral resection of the tumor), extirpation of the eye (enucleation), or exenteration of the orbit [70]. Brachytherapy is one of the most commonly used treatment modalities in choroidal melanoma. In this method, a radioactive plaque/applicator containing a radioactive element is sewn onto the sclera at the tumor base. The size and shape of the plaque are adjusted to the diameter of the tumor base, corrected for a normal tissue margin of at least 2 mm. Nowadays, brachytherapy is most often carried out with iodine (I-125) or ruthenium (Ru-106) plaques. Ruthenium emits beta radiation and finds application in the case of tumors with a thickness no greater than 5–6 mm. In turn, radioactive iodine emits gamma radiation, and thus is used for the irradiation of thicker tumors. The dose of radiation delivered to the apex and base of the tumor is computed adequately to the lesion’s diameter and applicator’s radioactivity. Brachytherapy destroys the DNA of melanoma cells and ultimately leads to the destruction of the tumor. Additionally, irradiation of the tumor causes fibrosis and occlusion of its intrinsic vessels [73].

Another treatment option available nowadays is irradiation of melanoma with high doses of energy originating from charged elementary particles, protons, or helium ions. Moreover, melanomas can be treated with a diode laser with 810 nm wavelength, a technique known as transpupillary thermotherapy. The laser beam heats the tumor with resultant hypothermia of up to 45–60 °C. The treatment causes direct thermal injury to melanoma cells, destroys tumor vessels, and disrupts the activity of cellular enzymes. The necrosis of tumor tissue reaches 3–4 mm in depth. Thermotherapy has optimal outcomes when combined with brachytherapy, the so-called sandwich method. In the case of small melanomas, brachytherapy can be followed by photodynamic therapy, with indocyanine green accumulating in tumor tissue and participating in the photodynamic reaction as a photosensitizing agent [74].

Whenever the tumor is not suitable for conservative treatment, it needs to be treated surgically by resection or enucleation of the eye in the case of more advanced lesions. If the tumor grows beyond the bulb, exenteration of the orbit is an option. Although surgical treatments of choroidal melanoma (pars plana vitrectomy or external incision) raise controversies, as they may facilitate the systemic spread of the tumor. However, the COMS report 28 found that there were no survival differences between patients whose primary choroidal tumors were treated with brachytherapy and those treated by enucleation [73].

Monitoring of treatment outcomes and tumor regression plays a significant role in evaluating therapy effectiveness. Irradiation, still the most commonly used treatment modality, leads to tumor shrinkage, scarification, and reduction in vasculature and retinochoroidal atrophy around the lesion’s base. OCT is an optimal instrument to visualize those changes. Together with US, OCT is the only effective method to estimate the size of the shrinking tumor. The measurements need to be accurate. Cennamo et al. [75] analyzed choroidal melanomas 1 year after irradiation with Ru-106 and documented a decrease in tumor size by 34.2% of average, with mean tumor basal diameter reduced from 7.88 mm before brachytherapy to 5.19 mm after the irradiation. OCTA is also suitable for the accurate measurement of tumor vasculature before and after treatment. In the study mentioned above, irradiation with Ru-106 contributed to a decrease in mean tumor vessel area from 11.61 mm^2^ to 9.16 mm^2^ and caused a reduction in mean tumor flow area from 6.05 mm^2^ to 4.09 mm^2^.

OCTA scans demonstrate a substantial decrease in vascularity density within irradiated melanomas, along with a reduction in tumor vessel diameters, already 3–6 months after the treatment [76]. Control OCTA scans obtained at regular intervals after irradiation allow for monitoring tumor regression and identifying changes suggesting potential recurrence. Partial or complete reduction in intrinsic tumor vascularity constitutes reliable proof of its regression, whereas neovascularization is a marker of recurrence [77]. Aside from OCTA, tumor vascularity can be also assessed on ICGA and (less effectively) on FA; however, the latter two methods are invasive and, as such cannot be used too often. Under such circumstances, OCTA plays a pivotal role as an instrument for regular, frequent monitoring of treatment outcomes in choroidal melanoma.

OCTA, conducted after a mean time of 46 months after I-125 plaque radiotherapy, demonstrates a lack of perfusion in the SCP and DCP in 65.3% of eyes and the loss of choriocapillaris within tumor margins in 65.2%. Qualitative assessment has demonstrated that irradiated eyes have significantly larger FAZ in the superficial plexus (0.961 mm^2^) and the deep plexus (1.396 mm^2^) than the contralateral, non-irradiated (control) eyes (0.280 mm^2^ and 0.458 mm^2^, respectively) [78]. A similar study involving patients after brachytherapy with Ru-106 also demonstrated that both sFAZ and dFAZ in the irradiated eyes were larger than in the control eyes (1629 µm^2^ vs. 428 µm^2^ and 1837 µm^2^ vs. 268 µm^2^, respectively) [79]. A correlation was observed between dFAZ and best-corrected visual acuity (BCVA): the larger the dFAZ, the worse the prognosis in terms of visual acuity. Foveal and parafoveal sCVD and dCVD were shown to be reduced in all irradiated eyes [78,79]. CVD was significantly lower in all sectors (superior, inferior, temporal, and nasal) of the posterior pole [74]. A strong positive correlation was found between the radiation dose and reduction in vascular density. The reduction in vascular density was the most evident in sectors located near the radioactive plaque, where the radiation dose was the highest [80]. Vascular loss is particularly evident in eyes that receive radiation doses higher than 45 Gy to the fovea [81].

Aside from documenting changes within the tumor, SCP, DCP, and choriocapillaris, OCT is also a basic examination in evaluating radiation maculopathy. Clinical features of evident radiation maculopathy include retinal exudation, hemorrhage, edema, presence of retinal telangiectasia, microaneurysms, and neovascularization [82]. In turn, characteristic features of radiation maculopathy on OCTA scans include the presence of microaneurysms, cysts, vascular dilations at the level of the SCP and DCP, FAZ disruption, and capillary dropout. According to the literature, radiation maculopathy may develop in 62–80.6% of irradiated eyes, and its presence is associated with a substantial loss of visual acuity [76,78,83]. A decrease in vascular density within the SCP and DCP can be detected early on OCTA, even before clinical features of maculopathy can be observed on FA and OCT. The changes in the DCP turned out to be more evident than those in SCP. Also, after proton-beam therapy, vascular density was shown to be reduced more in the DCP than in the SCP [83]. The prevalence of characteristic findings in the SCP and DCP differs too: while capillary loss is observed in 100% of each, microaneurysms are found in 65% and 76% of the SCP and DCP specimens, respectively, vascular dilation in 47% and 59%, respectively, and numerous cysts in 18% and 53%, respectively [83]. It is postulated that more deeply located capillaries of the DCP, acting as terminal vessels, are thinner, and hence more sensitive to irradiation than the larger capillaries of the SCP. Microscopic studies have demonstrated that irradiation leads to endothelial cell loss, with resultant occlusion of the capillaries. Small terminal capillaries are also more sensitive to ischemic stress [84]. Recent evidence suggests that a decrease in the vascular density of the DCP in foveal and parafoveal areas is the first sign of developing radiation maculopathy and can be considered a biomarker thereof [79]. OCTA is the only diagnostic instrument capable of capturing those changes. Based on increased CMT and cyst findings on OCTA and ophthalmoscopy, a new scoring system, from 0 to 5, was developed for the severity of radiation maculopathy, where 0 corresponds to the lack of maculopathy and 5 to its most severe form [85].

Prevention or treatment of the radiation-induced damage to retinal vasculature in patients treated for choroidal melanoma includes intravitreal administration of anti-vascular endothelial growth factor (anti-VEGF). Early administration of anti-VEGF can prevent fully symptomatic maculopathy and irreversible vision loss [86]. OCTA is the main instrument for the early detection and monitoring of subclinical forms of radiation maculopathy and identifying patients who might benefit from the anti-VERGF injection. Evaluation of anti-VEGF treatment outcomes with conbercept has demonstrated an increase in sCVD [87]. The underlying mechanism of the increase in blood-flow density after administering the anti-VEGF agent is yet to be understood. The timing of anti-VEGF administration is important, as this agent should be given after completing radiotherapy, when the tumor remains inactive. This prevents a potential decrease in the radiosensitivity of the tumor, associated with ischemia [87].

Despite their unquestioned advantages, OCT and OCTA do not always provide images of ideal quality, and some artifacts may occur and hinder the interpretation of the results. The occurrence of artifacts is an inevitable feature of all imaging techniques and may be related to the device’s functioning or patient behavior. To reduce such artifacts, software and hardware applications have been developed, along with hardware modifications, such as high-speed eye-tracking systems or implementation of ultrahigh scanning speeds. Research has shown that artifacts occur significantly more often in eyes with poor visual acuity and post-plaque radiotherapy radiation maculopathy than in normal eyes. Potential artifacts included specular dots, blink lines, motion artifacts, loss of signal, loss of focus, and edge duplication. At least one artifact was found in up to 94% of melanoma eyes and only 54% of normal eyes [88]. The most commonly found artifacts were loss of focus, blink lines, and motion artifacts. During the evaluation of capillary density and FAZ, blink lines may be misinterpreted as low flow or even non-perfusion. In turn, motion artifacts, loss of focus, and edge duplication may suggest increased capillary flow and vascularity. Thus, poor quality of OCTA scans may result in incorrect interpretation of the results.

## 9. Conclusions

(1)EDI-OCT is a state-of-the-art, non-invasive, most effective diagnostic method to detect small choroidal melanomas.(2)Incorporating OCT in the evaluation of choroidal melanoma allows for the visualization of anatomical details of the retina and choroid, which are characteristic for this malignancy and cannot be demonstrated with other techniques.(3)OCT provides accurate tumor-size measurements, also in the case of small choroidal melanomas and flat nevi, which are often not visible on conventional ultrasound.(4)Ultrasonographic measurements of tumors are biased and overestimated in relation to OCT-based measurements.(5)EDI-OCT demonstrates characteristic changes in the microstructure of individual layers of the retina and choroid, which facilitates differentiating between small choroidal melanomas and choroidal nevi.(6)OCT can detect characteristic clinical features differentiating choroidal melanoma from metastases to the choroid.(7)OCTA is an excellent, non-invasive diagnostic instrument to evaluate the intrinsic vasculature of choroidal melanoma, consisting of high-density vessels with variable thickness and irregular course, forming multiple turns and loops.(8)OCT and OCTA can be used to monitor changes suggesting the malignant transformation of a choroidal nevus to melanoma; because of the non-invasive nature of these examinations, they can be repeated as often as needed.(9)OCT and OCTA are useful diagnostic tools to monitor treatment outcomes in choroidal melanoma.

## Figures and Tables

**Table 1 cancers-14-03354-t001:** Features of various optical coherence tomography systems.

Features	TD-OCT	SD-OCT	SS-OCT
	Movable Reference Mirror	Mobile Reference Mirror, Spectrometer	Swept-Source Laser
Wavelength	810 nm	840 nm	1052 nm
Axial resolution	10 µm	8 µm	6 µm
Lateral resolution	20 µm	20 µm	20 µm
Speed of A scan per sec	512	50,000	100,000
B scan measurement time	-	1.0 s (50×)	1.0 s (96×)
Length of line scan	6 mm	Up to 9 mm	Up to 12 mm
Artifacts	more	less	less

**Table 2 cancers-14-03354-t002:** Clinical characteristics of T1–T4 choroidal melanomas.

Category	T1	T2	T3	T4
Incidence (% from 100%)	46	27	21	6
Age at presentation (years)	57	58	58	61
Tumor base (mm)	8	12	15	20
Tumor thickness (mm)	3.5	5.2	8.9	11.4
Mushroom-shaped (%)	8	20	38	39
Associated subretinal fluid (%)	64	80	82	83
Intraocular hemorrhage (%)	5	12	17	18
Rupture of Bruch’s membrane (%)	9	24	40	40
Extraocular extension (%)	<1	1	4	12
Metastases at 10 years (%)	15	25	49	63

Source: Reference [7].

## Data Availability

All materials and information will be available upon email request to the corresponding author.

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
