# Peer review of "Importance of Optical Coherence Tomography and Optical Coherence Tomography Angiography in the Imaging and Differentiation of Choroidal Melanoma: A Review"

_cancers, 2022, doi:10.3390/cancers14143354_

Round 1

Reviewer 1 Report

This paper reviews the development of OCT/OCTA and their applications in imaging vascular layers of the eye and in differential diagnosis of choroidal malignant lesions vs nevi and metastases.

Comments:

- The manuscript was not submitted with numbered lines, making it difficult to indicate precisely which lines correspond to the reviewer’s comments.

- The title is incomplete: correct for “[…] Choroidal Melanoma: A Systematic Review”.

- No figure or schematic drawing is included in the review. It might have helped non-visual science readers to include figures regarding the appearance of ocular tissues (retina/choroid) and tumors using different kinds of OCT/OCTA imaging technologies.

- Section 1 Introduction: This paragraph is missing references.

- Section 1 Introduction: I disagree with the following sentence in is actual form since FNAB is used routinely in clinical practice in North America and Europe to perform genetic testing for diagnosis/prognosis purposes. “Fine needle biopsy plays a minor role in evaluating choroidal melanoma; although the biopsy is suitable for establishing the ultimate diagnosis, it is rarely used in clinical practice given the potential risk of cancer cell spread and retinal damage”. The authors did not include references to support this statement. 

- Section 2, 2nd paragraph: Modify for “it utilizes light waves instead of high-frequency sound waves”.

- Section 2, 5th paragraph: Is “resignation” the correct term?

- Section 2, 7th paragraph: Correct for “two distant structures as corpus vitreum and choroid”.

- Section 2, 8th paragraph: Modify for “This UHR-OCT technology is currently used”.

- Section 5, 2nd paragraph: I disagree with the following sentence since there is a low annual rate (1/8,845) of malignant transformation of choroidal nevi in the white population (PMID: 16154197). “It is assumed that all choroidal melanomas originate from preexisting choroidal nevi”. The authors did not include references to support this statement. 

- Section 8, 1st paragraph: Correct for “a radioactive element is sewn onto the sclera”.

- Section 8, 3rd paragraph: I disagree with the following sentence since the COMS report no.28 found that there are no survival differences between patients whose primary choroidal tumors were treated with brachytherapy and those treated by enucleation (PMID: 17159027). “However, surgical treatments of choroidal melanoma raise controversies, as they may facilitate the systemic spread of the tumor”. The authors did not include references to support this statement. 

Author Response

Reviewer 1

     We would like to thank you for the detailed review of our manuscript and your valuable remarks. The manuscript has been rechecked and the necessary changes have been made in accordance with the Reviewers’ suggestions. The responses to all comments have been prepared and given below. We hope that you will find our explanations and manuscript modifications satisfactory for reconsidering its publication in Cancers. We highlighted the changes with the red font.

Comment 1

 The manuscript was not submitted with numbered lines, making it difficult to indicate precisely which lines correspond to the reviewer’s comments.

Response

We are very sorry or this overlooking. The numbers of lines were added.

Comment 2

- The title is incomplete: correct for “[…] Choroidal Melanoma: A Systematic Review”.

Response

We are sorry for copy-paste mistake. The title was corrected.

Comment 3

- No figure or schematic drawing is included in the review. It might have helped non-visual science readers to include figures regarding the appearance of ocular tissues (retina/choroid) and tumors using different kinds of OCT/OCTA imaging technologies.

Response

Thank you for this comment. We agree with the Reviewer that pictures or images would make this paper more interesting, however this is a review paper not an original research and we would have to ask the authors of cited papers for permission for publication of their pictures. Instead of images we have added the Table 1: Comparing features of various optical coherence tomography systems (line 189)

Comment 4

- Section 1 Introduction: This paragraph is missing references.

Response

We are sorry for that overlooking. The citations have been added to the introduction section.

Comment 5

- Section 1 Introduction: I disagree with the following sentence in is actual form since FNAB is used routinely in clinical practice in North America and Europe to perform genetic testing for diagnosis/prognosis purposes. “Fine needle biopsy plays a minor role in evaluating choroidal melanoma; although the biopsy is suitable for establishing the ultimate diagnosis, it is rarely used in clinical practice given the potential risk of cancer cell spread and retinal damage”. The authors did not include references to support this statement. 

Response

We are sorry for generalization and being imprecise. We have rephrased this sentence as follows: Fine needle biopsy (FNAB) is used routinely in clinical practice to perform genetic testing for diagnosis/prognosis purposes and it is suitable for establishing the ultimate diagnosis, although it rises the potential risk of cancer cell spread and retinal damage [1]. And we have added the appropriate reference. (lines 25-28)

Comment 6

Section 2, 2nd paragraph: Modify for “it utilizes light waves instead of high-frequency sound waves”.

Response

The correction was made according to the Reviewer’s suggestion (line 77)

Comment 7

- Section 2, 5th paragraph: Is “resignation” the correct term?

Response

We have rephrased this sentence as follows: Due to the use of longer waves and replacement the sensitivity-limiting spectrometer for a photodetector, SS-OCT scanners have many advantages, such as higher scanning speed, reduced sensitivity loss, better penetration of light and capturing of signal from deeper layers, and denser scan patterns in a larger scanning area [8].(lines 107-110)

Comment 8

- Section 2, 7th paragraph: Correct for “two distant structures as corpus vitreum and choroid”.

Response

Thank you for this comment, the change was made accordingly (line 148)

Comment 9

- Section 2, 8th paragraph: Modify for “This UHR-OCT technology is currently used”.

Response

Thank you for this comment, the change was made accordingly (line 152)

Comment 10

- Section 5, 2nd paragraph: I disagree with the following sentence since there is a low annual rate (1/8,845) of malignant transformation of choroidal nevi in the white population (PMID: 16154197). “It is assumed that all choroidal melanomas originate from preexisting choroidal nevi”. The authors did not include references to support this statement. 

Response

Thank you for this constructive comment. We rephrased this sentence as follows: Annual rate of malignant transformation of choroidal nevi in the white population accounts for 1/8.845 [51], and we have added the abovementioned paper to the references section. (lines 390-392).

Comment 11

- Section 8, 1st paragraph: Correct for “a radioactive element is sewn onto the sclera”.

Response

Thank you for this comment, the change was made accordingly (line 617)

Comment 12

Section 8, 3rd paragraph: I disagree with the following sentence since the COMS report no.28 found that there are no survival differences between patients whose primary choroidal tumors were treated with brachytherapy and those treated by enucleation (PMID: 17159027). “However, surgical treatments of choroidal melanoma raise controversies, as they may facilitate the systemic spread of the tumor”. The authors did not include references to support this statement. 

Response

Thank you for this comment. We are sorry for misleading. By the “surgical treatment we meant pars plana vitrectomy or external incision and we do agree with the Reviewer. For clarification sake we rephrased this sentence as follows:

Although surgical treatments of choroidal melanoma (pars plana vitrectomy or external incision) raise controversies, as they may facilitate the systemic spread of the tumor. However, the COMS report no.28 found that there were no survival differences between patients whose primary choroidal tumors were treated with brachytherapy and those treated by enucleation [73]. (lines 641-646) and we have added the abovementioned paper to the references section.

Reviewer 2 Report

In this work, the authors present a review on the use of OCT and OCT-A imaging to differentiate choroidal melanoma.

Because of my field of expertise, I will address only the imaging technique.

In the introduction, the authors (AAs) refer to the “…relatively newly implemented technique…” OCT is now on the market for 30 years. Adjectives like new or recent no longer apply.

In section 2, the authors state that “The source of light in OCT is a superluminescent diode with a wavelength of 820–840 nm.” Indeed, while the initial OCTs made use of superluminescent diodes, nowadays, we can find OCT systems based on tunable laser sources, therefore not using a superluminescent diode and distinct wavelength ranges from the one presented. I think the work would benefit if a more global description were provided instead.

Also in section 2, the AAs state that “…high transverse resolution (number of A-scans per image in B-projection), typically 18–20 μm.” While common, there’s an error on the statement. For an imaging system, the resolution is (1/dor), with “dor” the “distance of resolution”. “dor” is the minimum distance between two objects to allow them to be perceived by the imaging system as two distinct objects. For a shorter separation between the two objects, these will be perceived in the image as only one. As such, transverse resolution is not related by any means to the number of A-scans composing a B-scan. The number of A-scans per B-scan is the sampling. A higher sampling implies a higher number of A-scans, irrespectively of the transversal resolution.

Still in section 2, the AAs state that “… using a light source which is partially coherent with detection in the optical frequency domain rather than in the time domain.” This sentence is not clear to me. Maybe re-phase it to “… using a partially coherent light source with detection in the optical frequency domain rather than in the time domain.”?

Also, the sentence “(2) optical tomography scanners using lasers as a source of light and a photodetector instead of the spectrometer, the so-called swept-source OCT (SS-OCT) devices.” is not clear. Maybe referring to the system as using a tunable laser source and a photodetector would make it clearer.

Yet in section 2, the AAs state that “An algorithmic modification of SS-OCT contributed to reducing the granular noise signal and enabled obtaining the so-called “speckle noise-free” images [9].” SS-OCT refers to a technique in which the laser source outputs a “single” wavelength at a time. Hence, referring to an “algorithmic modification” does not make sense, even though the statement is in line with the sentence in ref 9. Also, this reference points to another two that further point to yet another one, “Three-dimensional speckle reduction in optical coherence tomography through structural guided filtering.” (https://doi.org/10.1117/1.OE.53.7.073105). In this last paper, no single reference is made to the SS-OCT technique. Indeed, it refers to an image filtering technique that can be applied to any volume of data in which speckle noise is present. A better description and correct references should be used.

The paragraph (section 2) “Despite many advantages, ... which limits their application in clinical practice.” requires a reference supporting the statement.

Author Response

Reviewer 2

     We would like to thank you for the detailed review of our manuscript and your valuable remarks. The manuscript has been rechecked and the necessary changes have been made in accordance with the Reviewers’ suggestions. The responses to all comments have been prepared and given below. We hope that you will find our explanations and manuscript modifications satisfactory for reconsidering its publication in Cancers. We highlighted the changes with the red font.

In this work, the authors present a review on the use of OCT and OCT-A imaging to differentiate choroidal melanoma.

Because of my field of expertise, I will address only the imaging technique.

Comment 1

In the introduction, the authors (AAs) refer to the “…relatively newly implemented technique…” OCT is now on the market for 30 years. Adjectives like new or recent no longer apply.

Response

Thank you for this comment. We do agree with the Reviewer. We have changed this statement accordingly: In this context, implemented thirty years ago, non-invasive optical coherence tomography (OCT)…, (line 32)

Comment 2

In section 2, the authors state that “The source of light in OCT is a superluminescent diode with a wavelength of 820–840 nm.” Indeed, while the initial OCTs made use of superluminescent diodes, nowadays, we can find OCT systems based on tunable laser sources, therefore not using a superluminescent diode and distinct wavelength ranges from the one presented. I think the work would benefit if a more global description were provided instead.

Response

Thank you for this comment. We have changed this statement as follows: In the initial technique the source of light in OCT was a superluminescent diode with a wavelength of 820–840 nm, nowadays, we can find OCT systems based on tunable laser sources as well. (lines 71-73)

Comment 3

Also, in section 2, the AAs state that “…high transverse resolution (number of A-scans per image in B-projection), typically 18–20 μm.” While common, there’s an error on the statement. For an imaging system, the resolution is (1/dor), with “dor” the “distance of resolution”. “dor” is the minimum distance between two objects to allow them to be perceived by the imaging system as two distinct objects. For a shorter separation between the two objects, these will be perceived in the image as only one. As such, transverse resolution is not related by any means to the number of A-scans composing a B-scan. The number of A-scans per B-scan is the sampling. A higher sampling implies a higher number of A-scans, irrespectively of the transversal resolution.

 Response

Thank you very much for clarification of this subject. To avoid misleading we have deleted this statement.

Comment 4

Still in section 2, the AAs state that “… using a light source which is partially coherent with detection in the optical frequency domain rather than in the time domain.” This sentence is not clear to me. Maybe re-phase it to “… using a partially coherent light source with detection in the optical frequency domain rather than in the time domain.”?

Response

Thank you for this comment. We have changed this statement as follows: using a partially coherent light source with detection in the optical frequency domain rather than in the time domain (107-109)

Comment 5 

Also, the sentence “(2) optical tomography scanners using lasers as a source of light and a photodetector instead of the spectrometer, the so-called swept-source OCT (SS-OCT) devices.” is not clear. Maybe referring to the system as using a tunable laser source and a photodetector would make it clearer.

Response

Thank you for this comment. We have changed this statement as follows:

…and (2) optical tomography scanners using a tunable laser source and a photodetector instead (lines 113)

Comment 6

 Yet in section 2, the AAs state that “An algorithmic modification of SS-OCT contributed to reducing the granular noise signal and enabled obtaining the so-called “speckle noise-free” images [9].” SS-OCT refers to a technique in which the laser source outputs a “single” wavelength at a time. Hence, referring to an “algorithmic modification” does not make sense, even though the statement is in line with the sentence in ref 9. Also, this reference points to another two that further point to yet another one, “Three-dimensional speckle reduction in optical coherence tomography through structural guided filtering.” (https://doi.org/10.1117/1.OE.53.7.073105). In this last paper, no single reference is made to the SS-OCT technique. Indeed, it refers to an image filtering technique that can be applied to any volume of data in which speckle noise is present. A better description and correct references should be used.

The paragraph (section 2) “Despite many advantages, ... which limits their application in clinical practice.” requires a reference supporting the statement. The paragraph (section 2) “Despite many advantages, ... which limits their application in clinical practice.” requires a reference supporting the statement.

Response

We are sorry for incorrect referencing. We have reordered the references numbers and added the paper Cyrill Gyger, Roger Cattin, Pascal W. Hasler, and Peter Maloca "Three-dimensional speckle reduction in optical coherence tomography through structural guided filtering," Optical Engineering 53(7), 073105 (23 July 2014) to our references section and support our statement by appropriate reference.

Reviewer 3 Report

In this article, the authors reviewed the applications of the different image technologies in the detection and treatment of choroidal melanoma. They compared the advantages and limitations of these methods on different aspects, including resolution, depth penetration, scanning speed, image quality, artifacts and etc. The authors then summarized and suggested the applications for some of the imaging methods, including EDI-OCT, OCT, OCTA and so on.

This topic is academically and clinically inquisitive. However, this article lacks a comprehensive foundation of this topic and a good data presentation and visualization.

Major points:

1. The reviewer suggests the authors provide more details of the working principles of the major imaging methods discussed in this article. This would help readers better understand the applications, differences and limitations discussed.

2. For each section/application, some figures and tables to summarize and compare the features and drawbacks of these methods are highly recommended for a good data presentation and visualization. Similarly, figures and tables can also help to explain the working principles of these methods.

3. The authors pointed out that this review accounts for the “signal processing and display techniques”. However, the signal processing and display methods were only mentioned without too much details. 

Author Response

Reviewer 3

     We would like to thank you for the detailed review of our manuscript and your valuable remarks. The manuscript has been rechecked and the necessary changes have been made in accordance with the Reviewers’ suggestions. The responses to all comments have been prepared and given below. We hope that you will find our explanations and manuscript modifications satisfactory for reconsidering its publication in Cancers. We highlighted the changes with the red font.

In this article, the authors reviewed the applications of the different image technologies in the detection and treatment of choroidal melanoma. They compared the advantages and limitations of these methods on different aspects, including resolution, depth penetration, scanning speed, image quality, artifacts and etc. The authors then summarized and suggested the applications for some of the imaging methods, including EDI-OCT, OCT, OCTA and so on.

This topic is academically and clinically inquisitive. However, this article lacks a comprehensive foundation of this topic and a good data presentation and visualization.

Major points:

Comment 1.

 The reviewer suggests the authors provide more details of the working principles of the major imaging methods discussed in this article. This would help readers better understand the applications, differences and limitations discussed.

Response

Thank you for this suggestion. According to the Reviewer’s advice we have added the part about OCT technology and working principle  in section 2

Working principle of OCT is very similar to medical ultrasound technique, except that OCT uses light instead of sound. Light travels much faster than ultrasound (in a vacuum at 300 000 000 m/second, while sound travels at approximately 300 m/second). OCTs uses the rule of indirect low-coherence interferometry, in which a beam of light is concentrating onto the retina. Part of the light is directed to the sample and another portion is sent to a reference arm with a well-known length.  As a result, back-scattered light travels to a detector, which is compared to a reference beam of a known length to calculate the echo time delay of light. When the distance between the light source and retinal tissue equals the distance between the light source and reference mirror, the reflected light and the reference mirror interacts to produce an interference pattern. The interference measured by the photodetector is then converted to an A-scan signal. Axial imaging depth delimits the axial range which is covered in a B-Scan. It is defined by the maximum fringe frequency which can be identified, because maximum frequency of the interference spectrum decodes the maximum depth.

(lines 52-65)

  1. For each section/application, some figures and tables to summarize and compare the features and drawbacks of these methods are highly recommended for a good data presentation and visualization. Similarly, figures and tables can also help to explain the working principles of these methods.

Thank you for this suggestion. According to the Reviewer’s advice we have added the following Table in section 2

features

TD-OCT

SD-OCT

SS-OCT

movable reference mirror

mobile reference mirror, spectrometer

Swept source laser

wavelength

810 nm

840 nm

1052 nm

Axial resolution

10 µm

8 µm

6 µm

Lateral resolution

20 µm

20 µm

20 µm

Speed of A scan per sec

512

50,000

100,000

B scan measurement time

-

1.0 s (50x)

1.0 s (96 x)

Length of line scan

6 mm

Up to 9 mm

Up to 12 mm

Artefacts

more

less

less

  1. The authors pointed out that this review accounts for the “signal processing and display techniques”. However, the signal processing and display methods were only mentioned without too much details. 

We are sorry for misleading. Our intention was to create typically clinical paper which would be helpful for ophthalmologists who deal with choroidal melanoma. We did not want to focus on technical details because we are not an expert in technology. In our daily practice we deal in clinical diagnostic issue and we wanted to present the comprehensive data of research of choroidal melanoma diagnostic. For clarification sake we deleted this sentence from the abstract.  

Round 2

Reviewer 3 Report

The authors have responded to the reviewer’s major comments with sufficient details. This article could be accepted for publication.